# Health Equity and a Paradigm Shift in Occupational Safety and Health

**DOI:** 10.3390/ijerph19010349

**Published:** 2021-12-29

**Authors:** Michael A. Flynn, Pietra Check, Andrea L. Steege, Jacqueline M. Sivén, Laura N. Syron

**Affiliations:** Occupational Health Equity Program, National Institute for Occupational Safety and Health, 1090 Tusculum Ave., Cincinnati, OH 45226, USA; gfe8@cdc.gov (P.C.); avs0@cdc.gov (A.L.S.); qcp6@cdc.gov (J.M.S.); lwy9@cdc.gov (L.N.S.)

**Keywords:** occupational safety and health, health equity, social determinants of health, work, biopsychosocial model, inclusive research methods

## Abstract

Despite significant improvements in occupational safety and health (OSH) over the past 50 years, there remain persistent inequities in the burden of injuries and illnesses. In this commentary, the authors assert that addressing these inequities, along with challenges associated with the fundamental reorganization of work, will require a more holistic approach that accounts for the social contexts within which occupational injuries and illnesses occur. A biopsychosocial approach explores the dynamic, multidirectional interactions between biological phenomena, psychological factors, and social contexts, and can be a tool for both deeper understanding of the social determinants of health and advancing health equity. This commentary suggests that reducing inequities will require OSH to adopt the biopsychosocial paradigm. Practices in at least three key areas will need to adopt this shift. Research that explicitly examines occupational health inequities should do more to elucidate the effects of social arrangements and the interaction of work with other social determinants on work-related risks, exposures, and outcomes. OSH studies regardless of focus should incorporate inclusive methods for recruitment, data collection, and analysis to reflect societal diversity and account for differing experiences of social conditions. OSH researchers should work across disciplines to integrate work into the broader health equity research agenda.

## 1. Introduction

Increased levels of disease and poverty among workers during the industrial revolution led Rudolf Virchow and others to establish the field of social medicine, which explores how social and economic conditions affect health, disease, and the practice of medicine [1]. However, the field of occupational safety and health (OSH) has evolved over the past half-century from its historic roots in social medicine into a largely technical field that focuses on identifying and eliminating physical, chemical, biological, and ergonomic hazards found in the workplace [2,3]. Rooted in the biomedical model of health [4], OSH generally utilizes a reductionist approach to isolate and address single, proximate factors that “cause” an injury or illness. This model has led to significant improvements in worker health over the past 50 years [5]. Nevertheless, persistent inequities in the burden of occupational injuries and illnesses, as well as challenges associated with the fundamental reorganization of the world of work [6], highlight the need to expand the current paradigm to account for the social contexts within which occupational injuries and illnesses occur [7,8,9]. Consideration of the role that social institutions and norms play in the inequitable distribution of work-related risks and benefits across society, and resultant issues of health equity, are central to this shift in OSH from a biomedical to a biopsychosocial approach [4]. A biopsychosocial approach takes a more holistic view by exploring the dynamic, multidirectional interactions between biological phenomena, psychological factors, and social relationships and contexts, which constitute processes of human development over the life course.

While the biomedical model circumscribes most OSH research, it is important to recognize that the field has increasingly embraced research on health inequities, including a growing commitment over the past two decades from the National Institute for Occupational Safety and Health (NIOSH) to conduct and support health equity-focused research. For example, over the past five years the NIOSH Occupational Health Equity Program has worked to expand these research efforts and promote a biopsychosocial approach across the field. This article discusses a paradigm shift to a biopsychosocial approach and then describes in detail how this shift may impact three key areas of OSH: research that explicitly examines occupational health inequities, incorporation of inclusive methods across OSH research, and integration of work into health equity research.

## 2. A Paradigm Shift to Advance Health Equity

Central to the current approach to OSH is the biomedical model of medicine, which focuses on identifying a specific physical cause for illness or injury and eliminating it [4]. The field of epidemiology in general and the tools of OSH epidemiology and surveillance have become increasingly tied to this epistemology. This approach has contributed to significant declines in work-related illness and injury over the past 50 years [5]. There is growing recognition, however, of the need for a more holistic and nuanced perspective on work and its impact on population health [10,11,12]. The declines in worker illnesses and injuries have not been distributed evenly across worker populations. Factors such as growing social inequality (along the lines of race/ethnicity, gender, and other social axes), restructuring of employer-employee relationships, and subcontracting practices that externalize risk highlight the need to account for the impact of the wider social context on work-related health outcomes [13]. These challenges lead us toward a biopsychosocial approach (viewing health within a social context) in OSH.

The concept of social determinants of health (SDOH), or how the structuring of society impacts the health and well-being of individuals and populations, can be useful towards understanding the biopsychosocial model of health. Social structures are dynamic and are continually shaped and reshaped by the distribution of power, money, and resources embedded in the social, political, and economic organization of society [14,15]. Work itself is a social determinant that affects the distribution of injuries, illnesses, health and well-being in society [10]. Work is currently listed as contributing to two (employment stability and social and community context) of the five key domains of SDOH within the Healthy People 2030 framework [16]. Many social determinants shape the inequitable distribution of work-related health risks and benefits [12]. Social determinants of health also often interact and overlap with one another in ways that can further privilege or disadvantage individual workers [17].

Social structures influence more than just the distribution of health and safety exposures, risks, and outcomes. They also contribute to exclusionary research, prevention, and mitigation practices which are often inadvertently tailored to the normative group [18]. As a result, those most in need of benefiting from preventive interventions are often least likely to receive them [19,20]. Public health interventions that do not account for these structural limitations can actually aggravate inequities as they often disproportionally help members of socially privileged groups [21].

## 3. Work, Health Inequities, and Society

Not all workers have the same risk of experiencing a work-related injury or illness, even when they have the same job. The way societies configure social and economic institutions creates the social conditions that influence workers’ exposures to occupational hazards (differential exposure) and their abilities to cope with risks or adverse consequences of an occupational injury or illness (differential susceptibility) [22]. Occupational health inequities are avoidable differences in work-related injury and illness incidence, morbidity and mortality that are closely linked with social, economic, and environmental disadvantage resulting from social arrangements [23]. Perhaps the three most salient social determinants of worker health are structures of social groups, industries, and jobs. Workers from certain groups, such as racialized/ethnic minorities and immigrants, are sorted into and overrepresented in dangerous occupations [24,25], receive differential treatment on the job [26], and have limited access to worker protection resources and benefits [27,28]. Industry structures can favor the health and well-being of some workers over others through, for example, the competitive bidding process and practice of externalizing costs, risks, and liability from large corporations to smaller ones through the use of sub-contracting arrangements [13,29]. Similarly, non-standard work arrangements, shift work, and considerations of autonomy at work are just some of the ways jobs are structured that also impact the distribution of work-related benefits and risks [30,31,32]. Furthermore, work’s influence on health and illness goes beyond the specific conditions at work. Indeed, the structure of one’s job or career exerts a significant influence over other aspects of life that contribute or detract from an individual’s health and that of their family such as income, social status, housing, access to healthcare, and leisure time. While work is a social determinant that contributes to inequities, it can also be a principal mechanism for securing fundamental needs and increasing health equity and well-being [10,33,34,35].

## 4. Towards the Integration of a Biopsychosocial Approach to OSH Research

A biopsychosocial approach to OSH explicitly takes into account the role that social arrangements play in the inequitable distribution of work-related risks and benefits and can be a tool for both deeper understanding of the social determinants of health and advancing health equity [3]. The integration of a health equity perspective into OSH research requires organizational changes to:Promote research focused on identifying, understanding and ameliorating OSH-related inequities that are closely linked with social, economic, and environmental disadvantage;Integrate inclusive research practices across the OSH field so that the knowledge base reflects societal diversity and accounts for differing experiences of social conditions; andDevelop a better understanding of how work-related constructs contribute to the inequitable distribution of illness, injury, mortality, and wellbeing in society and how the world of work could be leveraged to improve community health.

### 4.1. Research Focused on Occupational Health Inequities

Traditional fields of study used in OSH, such as industrial hygiene, often utilize studies that test very specific exposure/disease associations via risk assessment tools and worksite analysis to evaluate potential physical, chemical, and biological risks. Epidemiologic research that focuses on health inequities allows us to distinguish the most salient determinants, industries, and worker populations for understanding inequities. While these studies are key for documenting population-level occupational health inequities, the analysis is often limited to a single or limited set of variables. Yet, since the social position of any individual is a complex, interwoven set of identities based on asymmetrical power relationships along social axes such as race/ethnicity, sex/gender, nativity, and class [36,37], there is a need for research on relationships and interactions between these factors, as well as more contextualized analyses. Increasing research that aims to understand the social and structural factors that influence OSH risks, exposures, and outcomes, and the relationships between these factors, is essential.

Workers from groups that are socially marginalized are often disproportionately exposed to structural disadvantages related to other social determinants of OSH. For example, immigrants and racialized/ethnic minorities are overrepresented in contingent work arrangements [38] and foreign-born workers are overrepresented in small construction firms and receive less training and less overall safety communication than those employed by large companies [27]. These overlapping structural vulnerabilities [17] result in what Gravel and Dubé have termed “cumulative precarity,” meaning social and structural factors interact to create risks greater than the sum of the risk from each individual factor [39]. Overlapping structural vulnerabilities, and the ways they create cumulative precarity, need to be systematically investigated to bring a more complete picture of how occupational health inequities operate [40].

In addition to identifying which social factors, individually or in combination, contribute to occupational health inequities, research is also needed to characterize how structural disadvantages materialize at the worksite and in the lives of workers. For example, in 2019, foreign-born individuals made up two-thirds of occupational fatalities among Latinos in the United States despite the fact that immigrants represent only about one third of the Latino population in the United States [41]. However, few studies explore factors related to the immigrant experience that may lead to increased work-related fatalities. One such study examines how the assignment of an “undocumented” status results in a complex web of economic and social consequences that places workers at increased risk for occupational injury or illness and limits their ability to address the risks they face at work [42]. Developing an understanding of what contributes to the distribution of risks across different worker groups, industries, and occupations (not solely in terms of individual worker characteristics, but rather a deep inquiry into the social and structural conditions that shape workers’ lives) is essential to creating effective strategies to reduce these inequities.

Intervention research needs to engage an equity lens. Efforts to address occupational health inequities generally attempt to integrate workers from disadvantaged groups into existing institutional practices and paradigms that focus on the individual worker within the context of the immediate workplace [8,43]. These efforts tend to emphasize improving workers’ safety knowledge and promoting behavioral modifications with workers, an approach that has limited success [44]. A small but growing literature documents efforts to culturally tailor interventions [28]. While these efforts often acknowledge social constraints that may contribute to increased risk, they generally focus on integrating creative and inclusive teaching methods. Safety knowledge and behavior modification often remain the focus of the intervention. Unfortunately, much of the work to address occupational health inequities often relies on simplistic, individualistic and uncritical models of culture that reinforce the predominant worldview that the workers’ ‘otherness’ is both the cause of inequalities and the target for interventions [45]. For example, commonplace descriptions of immigrant and racialized/ethnic minority workers as “hard-to-reach” suggests that it is something about “them” (i.e., language, culture, mistrust of institutions, and other factors) that limits the services they receive from OSH organizations. Framing the issue in this way hides the fact that safety and health institutions have evolved to better meet the needs of workers from the cultural majority or normative group more than those from other social groups. A first step in acknowledging and addressing these institutional limitations would be to change the conceptualization of these workers from “hard to reach” to “hardly reached”—in other words, asking not what makes these workers “hard to reach”, but rather asking what organizations need to do to develop the institutional capacity to more effectively work with an increasingly multicultural and diverse workforce [46].

### 4.2. Integrate Inclusive Research Practices across OSH

While not all OSH research needs to have a primary focus on health inequities, all research should account for the diversity in the workforce and the influence that exclusionary social structures have on work-related risks, exposures, and outcomes as well as on OSH research design itself. Currently, concerns over health inequities are largely the domain of individual standard bearers or specific programs within larger institutions. Integrating equity more thoroughly, through inclusive OSH research and interventions, will require converting individual concerns into a core institutional value and implementing practices that better address the realities of a diverse workforce. In short, this will require a shift in organizational culture from individual concern for health equity to institutionalized practice [47]. This transition not only addresses ethical concerns related to inclusion, but will make for better science, as designs and interpretations will take into better account real-world contexts and represent a broader range of worker experiences.

As mentioned above, the exclusionary social structures that operate within society at large also operate within OSH organizations [3]. As a result, the assumptions, practices and approaches that guide OSH research have, often inadvertently, evolved to more effectively serve the needs of workers from some social groups more than others [18,21]. Identifying and correcting these exclusionary assumptions, practices and approaches is essential to ensuring that OSH research is responsive to the diversity within the workforce and effectively accounts for the social context within which research and practice related to worker safety and health take place [48]. Identifying and enumerating all of the potential ways that social context may contribute to exclusionary research practices is a daunting task. To simplify, we have identified three key areas that may be a good place to begin.

#### 4.2.1. Structural Invisibility

As the old adage goes, “you can’t fix what you don’t see” [49]. Structural invisibility results from society’s privileging and paying attention to the experiences of some social groups over others [50]. One way this privileging occurs is through data collection and analysis. The principal way that OSH institutions “see” reality is through data. The limitations of OSH surveillance systems to collect demographic and other data relevant to the experiences of historically underrepresented groups such as racialized/ethnic minorities and immigrants is well documented [10,51,52]. Data on social factors that potentially contribute to the inequitable distribution of work-related benefits and risks are often not collected, analyzed or published, thus rendering occupational health inequities and the social conditions that shape them invisible to researchers and institutions. Some research practices that contribute to structural invisibility include: lack or superficial treatment of socioeconomic variables by data collection instruments [53]; not accurately collecting socioeconomic data [52]; exclusionary recruitment practices and underrepresentation of workers from minority groups in study samples [54]; and an absence of or inadequate data analysis plan to identify potential occupational health inequities [55,56].

Common analytical perspectives and practices developed within the dominant social structure reflect the myopic and reductionist approach of the biomedical model which “asks only biological questions about what are in fact biosocial phenomena” [14] (p. 1686). For example, standard practices in epidemiology such as interpreting race as an individual demographic characteristic rather than a social construct or statistically adjusting for race instead of investigating the root causes of racial inequalities can reinforce ideas of biologic determinism and reify them in the scientific literature [55,56]. The result is a decontextualization of occupational injuries and illnesses that hides the social drivers of the inequities which further, albeit erroneously, reinforces the biomedical paradigm. It also has real world consequences as the scientific results impact the scope and focus of future research as well as decisions on intervention resource allocation, thus perpetuating the inequities that have been rendered invisible. For example, statistically controlling for sex in analysis of data on urinary tract infections (UTI) implies that women’s higher rate of UTI is entirely attributable to anatomical difference [57]. The result is that social factors that may be contributing to these disparities, such as reduced access to restroom facilities for female employees relative to males, are made invisible. The real-world consequence is that the potential solution of increasing access to restroom facilities for female employees is overlooked. More sophisticated data collection and analytical approaches, rooted in the biopsychosocial paradigm, are required to create a fuller, more accurate picture in which the range of worker experiences can be visible.

#### 4.2.2. Institutionalized Exclusion

Central to the biomedical model is the scientific method, which is based on the belief that replicable experimentation results in objective, generalizable knowledge of biological processes which can be used to identify and eliminate injury and illness. Alternately, a biopsychosocial approach recognizes that scientific inquiry itself is socially constructed and its evolution has been circumscribed by the same exclusionary social structures (race, class, gender, nativity, and other factors) that result in occupational health inequities [9,58]. “Studying up” or turning the analytical gaze back on scientific inquiry and research practices allows us to identify how exclusionary social structures are often codified in research practices, instruments, and scientific models resulting in an inherent bias in favor of the normative group [59].

There is a long history of male bias in scientific modeling and data production [60]. For example, toxicology defines the standard human as a 70 kg male (definition has now been updated to 80 kg male), treating females simply as smaller males. This scientific practice clearly demonstrates a bias towards the normative group in US society (men) and as a result, toxicological research may not account for important biological differences between males and females. Similarly, personal protective equipment (PPE) has been designed based on anthropometric measurements taken from military recruits in the United States during the 1950s to 1970s, a sample that was largely male and white [61]. The result is a decrease in the ability to achieve good fits for PPE for women, people of color, and individuals with body sizes or shapes that do not conform to those of military recruits [62]. Bias in scientific models is not only carried over from legacies of the past but continues to be introduced today. For example, anecdotal evidence suggests that modeling and development of work-related exoskeletons are developed to fit the male form which can lead to poor comfort and low acceptance by women [63]; similar concerns have been raised around the integration of bias in the development of artificial intelligence [64]. Identifying exclusionary scientific research practices, instruments, and scientific models and finding ways to correct them is an essential, yet often overlooked, element of addressing occupational health inequities and ensuring that the benefits of OSH research are shared equally, by all.

#### 4.2.3. Unexamined Assumptions

Scientists may not be objective observers but rather actors that occupy social positions within society that influence how they perceive the world and their research [65]. Research is circumscribed by the social context within which it develops and is impacted by the cultural norms and biases of the scientists themselves [18]. These biases and norms may underlie all aspects of research studies, from what questions get asked and the methods used to answer them, to the interpretations of results and how a given study is presented to and received by research and practice communities [66,67,68]. Failure to recognize the impact of these social arrangements on research poses epistemological barriers that potentially affect the interpretation of data and construction of knowledge [69]. However, this is easier said than done. The perspectives and assumptions of the normative group are frequently perceived as commonsensical and universal rather than culturally bound, especially by members of that group [70]. These perspectives are socially sanctioned, normalized and empowered through institutions such as media, laws, education systems, and institutional practices, and often permeate research without attention or reflection [71]. Accounting for assumptions that result from one’s social position and disciplinary conventions is essential to conducting inclusive research [72]. Just as individual worker’s social positions are complex, dynamic, and interwoven sets of identities based on asymmetrical power relationships along social axes that balance privilege and exclusion, so too are the social positions and identities of researchers [37]. As a researcher, understanding one’s position within this complex social web and how that position circumscribes one’s perspective of the world and one’s approach to public health requires an interactive process of education, reflection and action [36,73].

Conceptual approaches and reflexive practices, such as cultural humility, help researchers recognize that they bring culturally bound assumptions to their work which need to be identified and made explicit [74,75,76]. Involving researchers and study partners from diverse social and disciplinary backgrounds and fostering a culture of inclusion that openly discusses these differences and their potential impacts on the research is essential to identifying and addressing unexamined assumptions within the research [77]. Reflecting on the positionality of those involved in the project is an essential step to engage in these discussions. For example, the authors have been able to draw on their intersectional identities belonging to non-normative groups such as women, LGBTQ people, black people, multiracial people, people from working-class family backgrounds, and people with family experiences as refugees and immigrants, to provide insights for this commentary into how society excludes the experiences of some individuals along multiple social axes. The authors’ experiences of privilege associated with their education levels, employment with the federal government, and identities as white, male, heterosexual, upper-class family background and native-born US citizens have influenced their awareness and perceptions of the dynamics of exclusion and privilege. This mix of personal experiences, together with their professional training and work experiences outlined in their biographies, contextualize the perspectives outlined in this paper.

Another way to make unexamined assumptions of the research team explicit is to incorporate practices and methods that can help identify assumptions in research designs and data collection instruments, and analysis, such as cognitive testing, when developing data collection instruments [78,79]. This is even true when using well-established, validated instruments [80]. Box 1 contains an example that highlights the importance of accounting for the assumptions of the research team. More robust approaches to identify and address the unexamined assumptions of researchers are needed to truly ensure research projects are inclusive of the workforce diversity and that the data collected accurately reflects the experience of all of the respondents.

Box 1Not All Perspectives are Created Equal.The following example comes from a larger study [81] of tuberculosis (TB) among Latino immigrant workers and has been simplified to highlight some of the key concepts in this section such as unexamined assumptions and socially endorsed perspectives. It should not be taken as a scientific reporting of the results of the individual interviews but rather as an example of how exclusionary practices can manifest themselves in practice.During a formative investigation of tuberculosis (TB) among Latino immigrant workers, participants were asked if the results of their TB test were positive, to which many answered “yes”. When asked if they were taking their medicine several answered “no”. According to the researchers’ understandings of the questions as they had written them, these results would have been interpreted as indicating that these workers had test results that indicated the presence of TB, that they required medication, and that they were noncompliant with their treatment.However, to account for any possible unexamined assumptions, a modified cognitive testing protocol was integrated into the interview. This additional step asked the respondents to explain to the researchers how they understood the key concepts in the interview. A common answer was that a “positive TB test result” was good news that meant they were not sick and therefore did not need medicine.While different, both the researchers’ and the respondents’ understanding of “positive test result” are reasonable, yet they are not treated equally by public health institutions and society in general. Had the respondents’ interpretations not been uncovered during cognitive testing, the researchers’ initial interpretations of these workers having TB and being noncompliant with treatment would have held as a result of the investigation. Had these findings been published they would have become reified in the scientific literature, further reinforcing the researchers’ perspective. The erroneous findings would not only have misrepresented the lives of these workers, but the results would potentially have influenced the focus and funding of future research and interventions on topics that were not addressing the real needs of this community.

### 4.3. The World of Work and Health Inequities

Socioecological models have long recognized that the impact of employment on health goes beyond conditions at work. Indeed, as described above, one’s job exerts a significant influence over other aspects of life that contribute to or detract from an individual’s health and that of their family [35,82,83]. However, the classification of exposures and outcomes as work-related or not often separates occupational health research and practice from the rest of public health and work-related variables are largely absent from health equity research [3,22]. Despite these limitations, there is a small but growing body of literature that explores work as a causal pathway of health inequities. For example, recent analyses show that higher education does not confer the same benefit of access to safe and higher quality jobs, with demonstrable inequities in late life cognitive function [84] and all-cause mortality [85]. In other words, education’s protective effects through occupation (i.e., higher education leading to better jobs and better health) differ by race and gender. Greater attention to the relationship between work and health by population health researchers is essential to improving our understanding of work as a social determinant of health inequities and its potential to mitigate them [34].

Since work has a significant impact on the ability to secure the basic needs that provide the foundation for health and well-being, it directly and indirectly intersects with and influences many of the other social determinants of health. Therefore, the world of work has powerful potential to be leveraged to address health inequities in general, not only those classified as “occupational.” At the societal level, labor policies and workforce development initiatives that improve the quality of jobs and increase access to “decent work” can be analyzed and understood through a public health lens [86]. Further research is needed to fully explicate the promise of public health impacts beyond the workplace of labor- and work-focused policies and interventions, such as job security, wage and hour laws, paid sick leave, and other factors [87]. At the organizational level, issues related to work design, such as contract, wage, hour, and benefit structures, can improve worker health and well-being on and off the job [88]. Innovative practices around, for example, safety culture, work stress, and work-life balance are already implemented by organizations. The biopsychosocial approach extends the way organizations consider the costs and benefits of these strategies, as the impacts likely interact with non-work factors to amplify results. Indeed, there is a growing body of literature that elaborates the conceptual rationale and explores the business case for expanding the breadth of workplace wellness programs to include a social determinants perspective [89]. The Total Worker Health^®^ framework is an effort to operationalize the broader conceptualization of the relationship between work and health so that it can be implemented and studied [90]. Conversely, the inclusion of work and social factors related to occupation in broader public health and health equity research and practice is vital to understanding and taking advantage of the intersection of work with other aspects of individuals’ and communities’ lives. Work’s potential as an intervention site to provide access to resources and improve social determinants of health is a powerful, yet underutilized tool, in addressing health inequities [33].

## 5. Conclusions

Health equity is a central element of a larger paradigm shift to a biopsychosocial approach to OSH. This shift requires a change in organizational culture that makes health equity an institutionalized element of practice aligned with organizational values rather than the domain of individual concern. While this shift does not require all research to focus on health equity, it does require all research to engage in inclusive methods that address concerns around structural invisibility, institutionalized exclusion and unexamined assumptions. How quickly and successfully OSH organizations adapt to this paradigm shift will largely depend on the institutional support given to this transition. Within the biomedical model that has dominated OSH over the past 50 years, research on the technical aspects of OSH has been privileged over research that explores its social aspects [3,22,91]. As a result, the social sciences are underrepresented in work on occupational safety and health and the field has developed a limited ability to account for the historical and social context that circumscribe the injury experience and contribute to elevated rates of injuries among workers from certain groups.

Integration of social scientists into occupational safety and health is essential to improving the depth, breadth and quality of research and interventions that address occupational health inequities. It is also a prerequisite for developing the institutional capacity to embrace a paradigm shift to the biopsychosocial approach. Successful integration of social scientists will require organizations to increase their internal capacity, expand external interest and foreground the social perspective. Perhaps the most commonsensical and effective approach to building institutional capacity in the social sciences will be to prioritize directly hiring professionals from underrepresented fields such as medical anthropology, health communications, sociology, social epidemiology, and translation research. Our own team for this paper represents some of the leaders in occupational health equity at the National Institute for Occupational Safety and Health, a federal government institution that is steeped in the normative culture of the United States and has extensive global reach and influence. Our training comes from diverse fields of anthropology, population health, and epidemiology and our experience covers surveillance, quantitative and qualitative methods, health communication, translation and intervention research, and public health programs and partnerships. Our work at NIOSH has cut across organizational lines as we have worked to promote health equity within our divisions and are moving as an Institute towards embracing the sorts of cultural shifts in norms, values, and practices described herein.

However, direct hires alone are not enough. Generating interest in OSH among nontraditional academic departments and professional organizations will be essential to improving occupational health equity research. It is easy to see how specific concerns around occupational health equity, such as gender inequity in exoskeleton design, racial bias in artificial intelligence, discrimination and workplace stress, alternative work arrangements and substance abuse, could be of interest to researchers in gender and ethnic studies, communications, anthropology and sociology, among others. In addition, an expanded framing of the relationship among work, health and inequity through sociocultural, biopsychosocial, and social determinants lenses not only makes OSH relevant to a larger number of academic fields but also leverages the awareness of these relationships that was built during the COVID-19 pandemic. Indeed, making explicit the implicit connections between public health, OSH, and the social sciences more broadly will go a long way in bridging the gap between OSH and the social sciences and improving our understanding of the social dimensions of worker health and well-being. Finally, ensuring social science perspectives influence the organizational direction, strategic plans, and budget decisions of OSH organizations is essential to promoting health equity research. Foregrounding a social perspective in OSH organizations will require the participation of social scientists in internal leadership positions as well as external influencers through their service on such bodies as advisory boards and grant review committees. The question left to these organizations is: How can we best leverage this moment to institutionalize a biopsychosocial approach to OSH?

## Data Availability

Not applicable.

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
