# Peer review of "Health Equity and a Paradigm Shift in Occupational Safety and Health"

_ijerph, 2021, doi:10.3390/ijerph19010349_

Round 1
Reviewer 1 Report
This is an excellent summary of these issues, and this commentary makes me want more details; maybe some will come in the research articles that come after in this IJERPH issue.
People not comfortable in social sciences may have a little trouble getting through this document. There is a lot of terminology that is pervasive in general health inequities literature, and some of these "catch phrases" could be difficult (put off?) some researchers in this field. At some point, a dedicated document could have boxes with concrete examples of all of these aspects described here. I am thinking of very practical displays of how assumptions and research methods hide inequities (there are several in the paper; I just think they could be made clearer/simpler in a different document, perhaps).
This focus connects with organizational psychology and the issue of safety culture could be described in this paper.
You cite NIOSH's strategic plan. One has to read between the lines (in the strat plan) in order to detect the viewpoint expressed in this commentary.
There are a couple typos on page 2 line 49: Health, conducting
Really, I have no negative comments. This is a great piece.
Author Response
Reviewer 1
1)This is an excellent summary of these issues, and this commentary makes me want more details; maybe some will come in the research articles that come after in this IJERPH issue.
We appreciate the positive feedback and hope to elaborate further on these issues in future articles.
2) People not comfortable in social sciences may have a little trouble getting through this document. There is a lot of terminology that is pervasive in general health inequities literature, and some of these "catch phrases" could be difficult (put off?) some researchers in this field. At some point, a dedicated document could have boxes with concrete examples of all of these aspects described here. I am thinking of very practical displays of how assumptions and research methods hide inequities (there are several in the paper; I just think they could be made clearer/simpler in a different document, perhaps).
In response to the recommendation of the Academic Editor we have included a pull-out box with one case study to improve the accessibility of the article to a non-specialist audience. We were not sure how to format the pull-out box, so we simply added the suggested text in italics (lines 341-370 in the Track Changes version).
3) This focus connects with organizational psychology and the issue of safety culture could be described in this paper.
The paper was revised to respond to this comment and additional language was included to the final section of the paper (lines 409 – 414 in the Track Changes version) that addresses key topics in organizational psychology such as safety culture, work stress, work-life balance. An additional citation to the NIOSH Health Work Design program was also included for those interested in more information on this topic.
4) You cite NIOSH's strategic plan. One has to read between the lines (in the strategic plan) in order to detect the viewpoint expressed in this commentary.
The text does not specifically mention the NIOSH Strategic Plan. We believe the reviewer is referring to our reference to the Occupational Health Equity Program Portfolio page (reference #23). This citation is included because it supports the definition of occupational health inequities referenced in the article.
5) There are a couple typos on page 2 line 49: Health, conducting
Thank you for catching these mistakes, the typos have been corrected
Reviewer 2 Report
The article is extremely interesting and inspiring. The biopsychosocial conception of OSH appears to be central to ensuring occupational health equity as a central element of a larger paradigm shift to a biopsychosocial approach to OSH. I consider the main added value of the article to be the definition of three key areas where the social context may contribute to exclusionary research practices: structural invisibility, institutionalized exclusion, unexamined assumptions.
I also find valuable the emphasis on the interdisciplinary dimension of research and the need to ensure that social science perspectives influence organisational direction, strategic plans and budgetary decisions of H&I organisations is essential to promote health equity research.
From my point of view, the only thing missing was an indication of the implications at the organisational level. In fact, occupational health equity practices are implemented in specific organisations and by specific employers, hence a paradigm shift and conceptualisation of OSH will have the greatest impact on them - in terms of costs and benefits. If the analysis of these two aspects is not attractive to them, then how do we ensure that the biosocial concept of OSH is implemented in specific workplaces?
The article is conceptual in nature, opening up the research. The authors based their analysis on existing research results and did not present the results of primary research, which is not indicated in the abstract and introduction. Hence, a slightly lower assessment of the methodological aspect - not for the lack of primary research, but for the lack of a clear statement of the nature of the article and the resulting research approach.
Author Response
Reviewer 2
- The article is extremely interesting and inspiring. The biopsychosocial conception of OSH appears to be central to ensuring occupational health equity as a central element of a larger paradigm shift to a biopsychosocial approach to OSH. I consider the main added value of the article to be the definition of three key areas where the social context may contribute to exclusionary research practices: structural invisibility, institutionalized exclusion, unexamined assumptions.
We appreciate the positive feedback and agree that the inclusive research section is one of the most important contributions the article makes.
- I also find valuable the emphasis on the interdisciplinary dimension of research and the need to ensure that social science perspectives influence organizational direction, strategic plans and budgetary decisions of H&I organizations is essential to promote health equity research.
We apricate the positive feedback and are glad to see these points resonated with the reviewer.
- From my point of view, the only thing missing was an indication of the implications at the organizational level. In fact, occupational health equity practices are implemented in specific organizations and by specific employers, hence a paradigm shift and conceptualization of OSH will have the greatest impact on them - in terms of costs and benefits. If the analysis of these two aspects is not attractive to them, then how do we ensure that the biosocial concept of OSH is implemented in specific workplaces?
Language was added (lines 409 – 417) that discusses the paradigm shift at the organizational level both conceptually and operationally. An addition citation (#88) was added to an article that discusses conceptual as well as practical aspects of the incorporation of a SDOH perspective by employers into worker wellness programs.
- The article is conceptual in nature, opening up the research. The authors based their analysis on existing research results and did not present the results of primary research, which is not indicated in the abstract and introduction. Hence, a slightly lower assessment of the methodological aspect - not for the lack of primary research, but for the lack of a clear statement of the nature of the article and the resulting research approach.
We have revised the abstract to state that this article is a commentary (lines 10 & 17).